# QUATERNION RECURRENT NEURAL NETWORKS

**Titouan Parcollet**[1,4], **Mirco Ravanelli**[2], **Mohamed Morchid**[1], **Georges Linarès**[1],
**Chiheb Trabelsi**[2,5], **Renato De Mori**[1,3], **Yoshua Bengio**[2] *

[1]LIA, Université d'Avignon, France
[2]MILA, Université de Montréal, Québec, Canada
[3]McGill University, Québec, Canada
[4]Orkis, Aix-en-provence, France
[5]Element AI, Montréal, Québec, Canada
`titouan.parcollet@alumni.univ-avignon.fr,`
`mirco.ravanelli@gmail.com,`
`firstname.lastname@univ-avignon.fr,`
`chiheb.trabelsi@polymtl.ca, rdemori@cs.mcgill.ca`

## ABSTRACT

Recurrent neural networks (RNNs) are powerful architectures to model sequential data, due to their capability to learn short and long-term dependencies between the basic elements of a sequence. Nonetheless, popular tasks such as speech or images recognition, involve multi-dimensional input features that are characterized by strong internal dependencies between the dimensions of the input vector. We propose a novel quaternion recurrent neural network (QRNN), alongside with a quaternion long-short term memory neural network (QLSTM), that take into account both the external relations and these internal structural dependencies with the quaternion algebra. Similarly to capsules, quaternions allow the QRNN to code internal dependencies by composing and processing multidimensional features as single entities, while the recurrent operation reveals correlations between the elements composing the sequence. We show that both QRNN and QLSTM achieve better performances than RNN and LSTM in a realistic application of automatic speech recognition. Finally, we show that QRNN and QLSTM reduce by a maximum factor of 3.3x the number of free parameters needed, compared to real-valued RNNs and LSTMs to reach better results, leading to a more compact representation of the relevant information.

## 1 INTRODUCTION

In the last few years, deep neural networks (DNN) have encountered a wide success in different domains due to their capability to learn highly complex input to output mapping. Among the different DNN-based models, the recurrent neural network (RNN) is well adapted to process sequential data. Indeed, RNNs build a vector of activations at each timestep to code latent relations between input vectors. Deep RNNs have been recently used to obtain hidden representations of speech unit sequences (Ravanelli et al., 2018a) or text word sequences (Conneau et al., 2018), and to achieve state-of-the-art performances in many speech recognition tasks (Graves et al., 2013a;b; Amodei et al., 2016; Povey et al., 2016; Chiu et al., 2018). However, many recent tasks based on multi-dimensional input features, such as pixels of an image, acoustic features, or orientations of 3D models, require to represent both external dependencies between different entities, and internal relations between the features that compose each entity. Moreover, RNN-based algorithms commonly require a huge number of parameters to represent sequential data in the hidden space.

Quaternions are hypercomplex numbers that contain a real and three separate imaginary components, perfectly fitting to 3 and 4 dimensional feature vectors, such as for image processing and robot kinematics (Sangwine, 1996; Pei & Cheng, 1999; Aspragathos & Dimitros, 1998). The

---

*CIFAR Senior Fellow

idea of bundling groups of numbers into separate entities is also exploited by the recent manifold and capsule networks (Chakraborty et al., 2018; Sabour et al., 2017). Contrary to traditional homogeneous representations, capsule and quaternion networks bundle sets of features together. Thereby, quaternion numbers allow neural network based models to code latent inter-dependencies between groups of input features during the learning process with fewer parameters than RNNs, by taking advantage of the *Hamilton product* as the equivalent of the ordinary product, but between quaternions. Early applications of quaternion-valued backpropagation algorithms (Arena et al., 1994; 1997) have efficiently solved quaternion functions approximation tasks. More recently, neural networks of complex and hypercomplex numbers have received an increasing attention (Hirose & Yoshida, 2012; Tygert et al., 2016; Danihelka et al., 2016; Wisdom et al., 2016), and some efforts have shown promising results in different applications. In particular, a deep quaternion network (Parcollet et al., 2016; 2017a;b), a deep quaternion convolutional network (Gaudet & Maida, 2018; Parcollet et al., 2018), or a deep complex convolutional network (Trabelsi et al., 2017) have been employed for challenging tasks such as images and language processing. However, these applications do not include recurrent neural networks with operations defined by the quaternion algebra.

This paper proposes to integrate local spectral features in a novel model called quaternion recurrent neural network[1] (QRNN), and its gated extension called quaternion long-short term memory neural network (QLSTM). The model is proposed along with a well-adapted parameters initialization and turned out to learn both inter- and intra-dependencies between multidimensional input features and the basic elements of a sequence with drastically fewer parameters (Section 3), making the approach more suitable for low-resource applications. The effectiveness of the proposed QRNN and QLSTM is evaluated on the realistic TIMIT phoneme recognition task (Section 4.2) that shows that both QRNN and QLSTM obtain better performances than RNNs and LSTMs with a best observed phoneme error rate (PER) of $18.5\%$ and $15.1\%$ for QRNN and QLSTM, compared to $19.0\%$ and $15.3\%$ for RNN and LSTM. Moreover, these results are obtained alongside with a reduction of 3.3 times of the number of free parameters. Similar results are observed with the larger Wall Street Journal (WSJ) dataset, whose detailed performances are reported in the Appendix 6.1.1.

## 2 MOTIVATIONS

A major challenge of current machine learning models is to well-represent in the latent space the astonishing amount of data available for recent tasks. For this purpose, a good model has to efficiently encode local relations within the input features, such as between the Red, Green, and Blue (R,G,B) channels of a single image pixel, as well as structural relations, such as those describing edges or shapes composed by groups of pixels. Moreover, in order to learn an adequate representation with the available set of training data and to avoid overfitting, it is convenient to conceive a neural architecture with the smallest number of parameters to be estimated. In the following, we detail the motivations to employ a quaternion-valued RNN instead of a real-valued one to code inter and intra features dependencies with fewer parameters.

As a first step, a better representation of multidimensional data has to be explored to naturally capture internal relations within the input features. For example, an efficient way to represent the information composing an image is to consider each pixel as being a whole entity of three strongly related elements, instead of a group of uni-dimensional elements that *could* be related to each other, as in traditional real-valued neural networks. Indeed, with a real-valued RNN, the latent relations between the RGB components of a given pixel are hardly coded in the latent space since the weight has to find out these relations among all the pixels composing the image. This problem is effectively solved by replacing real numbers with quaternion numbers. Indeed, quaternions are fourth dimensional and allow one to build and process entities made of up to four related features. The quaternion algebra and more precisely the *Hamilton product* allows quaternion neural network to capture these internal latent relations within the features encoded in a quaternion. It has been shown that QNNs are able to restore the spatial relations within 3D coordinates (Matsui et al., 2004), and within color pixels (Isokawa et al., 2003), while real-valued NN failed. This is easily explained by the fact that the quaternion-weight components are shared through multiple quaternion-input parts during the *Hamilton product* , creating relations within the elements. Indeed, Figure 1 shows that the multiple weights required to code latent relations within a feature are considered at the same level as for

---

[1] https://github.com/Orkis-Research/Pytorch-Quaternion-Neural-Networks

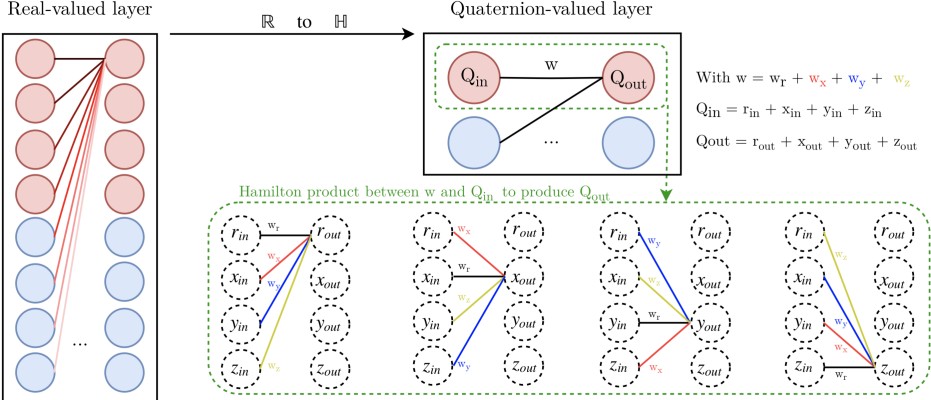

Figure 1: Illustration of the input features ($Q_{in}$) latent relations learning ability of a quaternion-valued layer (right) due to the quaternion weight sharing of the *Hamilton product* (Eq. 5), compared to a standard real-valued layer (left).

learning global relations between different features, while the quaternion weight $w$ codes these internal relations within a unique quaternion $Q_{out}$ during the *Hamilton product* (right).

Then, while bigger neural networks allow better performances, quaternion neural networks make it possible to deal with the same signal dimension but with four times less neural parameters. Indeed, a 4-number quaternion weight linking two 4-number quaternion units only has 4 degrees of freedom, whereas a standard neural net parametrization has $4 \times 4 = 16$, i.e., a 4-fold saving in memory. Therefore, the natural multidimensional representation of quaternions alongside with their ability to drastically reduce the number of parameters indicate that hyper-complex numbers are a better fit than real numbers to create more efficient models in multidimensional spaces. Based on the success of previous deep quaternion convolutional neural networks and smaller quaternion feed-forward architectures (Kusamichi et al., 2004; Isokawa et al., 2009; Parcollet et al., 2017a), this work proposes to adapt the representation of hyper-complex numbers to the capability of recurrent neural networks in a natural and efficient framework to multidimensional sequential tasks such as speech recognition.

Modern automatic speech recognition systems usually employ input sequences composed of multidimensional acoustic features, such as log Mel features, that are often enriched with their first, second and third time derivatives (Davis & Mermelstein, 1990; Furui, 1986), to integrate contextual information. In standard RNNs, static features are simply concatenated with their derivatives to form a large input vector, without effectively considering that signal derivatives represent different views of the same input. Nonetheless, it is crucial to consider that time derivatives of the spectral energy in a given frequency band at a specific time frame represent a special state of a time-frame, and are linearly correlated (Tokuda et al., 2003). Based on the above motivations and the results observed on previous works about quaternion neural networks, we hypothesize that quaternion RNNs naturally provide a more suitable representation of the input sequence, since these multiple views can be directly embedded in the multiple dimensions space of the quaternion, leading to better generalization.

## 3 QUATERNION RECURRENT NEURAL NETWORKS

This Section describes the quaternion algebra (Section 3.1), the internal quaternion representation (Section 3.2), the backpropagation through time (BPTT) for quaternions (Section 3.3.2), and proposes an adapted weight initialization to quaternion-valued neurons (Section 3.4).

### 3.1 QUATERNION ALGEBRA

The quaternion algebra $\mathbb{H}$ defines operations between quaternion numbers. A quaternion Q is an extension of a complex number defined in a four dimensional space as:

$$Q = r1 + x\mathbf{i} + y\mathbf{j} + z\mathbf{k}, \tag{1}$$

where $r$, $x$, $y$, and $z$ are real numbers, and 1, $\mathbf{i}$, $\mathbf{j}$, and $\mathbf{k}$ are the quaternion unit basis. In a quaternion, $r$ is the real part, while $x\mathbf{i} + y\mathbf{j} + z\mathbf{k}$ with $\mathbf{i}^2 = \mathbf{j}^2 = \mathbf{k}^2 = \mathbf{ijk} = -1$ is the imaginary part, or the vector part. Such a definition can be used to describe spatial rotations. The information embedded in the quaterion $Q$ can be summarized into the following matrix of real numbers:

$$Q_{mat} = \begin{bmatrix} r & -x & -y & -z \\ x & r & -z & y \\ y & z & r & -x \\ z & -y & x & r \end{bmatrix}. \tag{2}$$

The conjugate $Q^*$ of $Q$ is defined as:

$$Q^* = r1 - x\mathbf{i} - y\mathbf{j} - z\mathbf{k}. \tag{3}$$

Then, a normalized or unit quaternion $Q^{\triangleleft}$ is expressed as:

$$Q^{\triangleleft} = \frac{Q}{\sqrt{r^2 + x^2 + y^2 + z^2}}. \tag{4}$$

Finally, the *Hamilton product* $\otimes$ between two quaternions $Q_1$ and $Q_2$ is computed as follows:

$$\begin{aligned} Q_1 \otimes Q_2 = &(r_1 r_2 - x_1 x_2 - y_1 y_2 - z_1 z_2) + (r_1 x_2 + x_1 r_2 + y_1 z_2 - z_1 y_2)\boldsymbol{i} + \\ &(r_1 y_2 - x_1 z_2 + y_1 r_2 + z_1 x_2)\boldsymbol{j} + (r_1 z_2 + x_1 y_2 - y_1 x_2 + z_1 r_2)\boldsymbol{k}. \end{aligned} \tag{5}$$

The *Hamilton product* (a graphical view is depicted in Figure 1) is used in QRNNs to perform transformations of vectors representing quaternions, as well as scaling and interpolation between two rotations following a geodesic over a sphere in the $\mathbb{R}^3$ space as shown in (Minemoto et al., 2017).

## 3.2 QUATERNION REPRESENTATION

The QRNN is an extension of the real-valued (Medsker & Jain, 2001) and complex-valued (Hu & Wang, 2012; Song & Yam, 1998) recurrent neural networks to hypercomplex numbers. In a quaternion dense layer, all parameters are quaternions, including inputs, outputs, weights, and biases. The quaternion algebra is ensured by manipulating matrices of real numbers (Gaudet & Maida, 2018). Consequently, for each input vector of size $N$, output vector of size $M$, dimensions are split into four parts: the first one equals to $r$, the second is $x\mathbf{i}$, the third one equals to $y\mathbf{j}$, and the last one to $z\mathbf{k}$ to compose a quaternion $Q = r1 + x\mathbf{i} + y\mathbf{j} + z\mathbf{k}$. The inference process of a fully-connected layer is defined in the real-valued space by the dot product between an input vector and a real-valued $M \times N$ weight matrix. In a QRNN, this operation is replaced with the *Hamilton product* (Eq. 5) with quaternion-valued matrices (i.e. each entry in the weight matrix is a quaternion). The computational complexity of quaternion-valued models is discussed in Appendix 6.1.2

## 3.3 LEARNING ALGORITHM

The QRNN differs from the real-valued RNN in each learning sub-processes. Therefore, let $x_t$ be the input vector at timestep $t$, $h_t$ the hidden state, $W_{hx}$, $W_{hy}$ and $W_{hh}$ the input, output and hidden states weight matrices respectively. The vector $b_h$ is the bias of the hidden state and $p_t$, $y_t$ are the output and the expected target vectors. More details of the learning process and the parametrization are available on Appendix 6.2.

### 3.3.1 FORWARD PHASE

Based on the forward propagation of the real-valued RNN (Medsker & Jain, 2001), the QRNN forward equations are extended as follows:

$$h_t = \alpha(W_{hh} \otimes h_{t-1} + W_{hx} \otimes x_t + b_h), \tag{6}$$

where $\alpha$ is a *quaternion split activation function* (Xu et al., 2017; Tripathi, 2016) defined as:

$$\alpha(Q) = f(r) + f(x)\mathbf{i} + f(y)\mathbf{j} + f(z)\mathbf{k}, \tag{7}$$

with $f$ corresponding to any standard activation function. The split approach is preferred in this work due to better prior investigations, better stability (i.e. pure quaternion activation functions contain singularities), and simpler computations. The output vector $p_t$ is computed as:

$$p_t = \beta(W_{hy} \otimes h_t), \tag{8}$$

where $\beta$ is any split activation function. Finally, the objective function is a classical loss applied component-wise (e.g., mean squared error, negative log-likelihood).

### 3.3.2 QUATERNION BACKPROPAGATION THROUGH TIME

The backpropagation through time (BPTT) for quaternion numbers (QBPTT) is an extension of the standard quaternion backpropagation (Nitta, 1995), and its full derivation is available in Appendix 6.3. The gradient with respect to the loss $E_t$ is expressed for each weight matrix as $\Delta_{hy}^t = \frac{\partial E_t}{\partial W_{hy}}$, $\Delta_{hh}^t = \frac{\partial E_t}{\partial W_{hh}}$, $\Delta_{hx}^t = \frac{\partial E_t}{\partial W_{hx}}$, for the bias vector as $\Delta_b^t = \frac{\partial E_t}{\partial B_h}$, and is generalized to $\Delta^t = \frac{\partial E_t}{\partial W}$ with:

$$\frac{\partial E_t}{\partial W} = \frac{\partial E_t}{\partial W^r} + \mathbf{i}\frac{\partial E_t}{\partial W^i} + \mathbf{j}\frac{\partial E_t}{\partial W^j} + \mathbf{k}\frac{\partial E_t}{\partial W^k}. \tag{9}$$

Each term of the above relation is then computed by applying the chain rule. Indeed, and conversaly to real-valued backpropagation, QBPTT must defines the dynamic of the loss *w.r.t* to each component of the quaternion neural parameters. As a use-case for the equations, the mean squared error at a timestep $t$ and named $E_t$ is used as the loss function. Moreover, let $\lambda$ be a fixed learning rate. First, the weight matrix $W_{hy}$ is only seen in the equations of $p_t$. It is therefore straightforward to update each weight of $W_{hy}$ at timestep $t$ following:

$$W_{hy} = W_{hy} - \lambda\Delta_{hy}^t \otimes h_t^*, \text{ with } \Delta_{hy}^t = \frac{\partial E_t}{\partial W_{hy}} = (p_t - y_t), \tag{10}$$

where $h_t^*$ is the conjugate of $h_t$. Then, the weight matrices $W_{hh}$, $W_{hx}$ and biases $b_h$ are arguments of $h_t$ with $h_{t-1}$ involved, and the update equations are derived as:

$$W_{hh} = W_{hh} - \lambda\Delta_{hh}^t, \quad W_{hx} = W_{hx} - \lambda\Delta_{hx}^t, \quad b_h = b_h - \lambda\Delta_b^t, \tag{11}$$

with,

$$\Delta_{hh}^t = \sum_{m=0}^{t}(\prod_{n=m}^{t} \delta_n) \otimes h_{m-1}^*, \quad \Delta_{hx}^t = \sum_{m=0}^{t}(\prod_{n=m}^{t} \delta_n) \otimes x_m^*, \quad \Delta_b^t = \sum_{m=0}^{t}(\prod_{n=m}^{t} \delta_n), \tag{12}$$

and,

$$\delta_n = \begin{cases} W_{hh}^* \otimes \delta_{n+1} \times \alpha'(h_n^{preact}) & \text{if } n \neq t \\ W_{hy}^* \otimes (p_n - y_n) \times \beta'(p_n^{preact}) & \text{otherwise}, \end{cases} \tag{13}$$

with $h_n^{preact}$ and $p_n^{preact}$ the pre-activation values of $h_n$ and $p_n$ respectively.

### 3.4 PARAMETER INITIALIZATION

A well-designed parameter initialization scheme strongly impacts the efficiency of a DNN. An appropriate initialization, in fact, improves DNN convergence, reduces the risk of exploding or vanishing gradient, and often leads to a substantial performance improvement (Glorot & Bengio, 2010). It has been shown that the backpropagation through time algorithm of RNNs is degraded by an inappropriated parameter initialization (Sutskever et al., 2013). Moreover, an hyper-complex parameter cannot be simply initialized randomly and component-wise, due to the interactions between components. Therefore, this Section proposes a procedure reported in Algorithm 1 to initialize a matrix $W$ of quaternion-valued weights. The proposed initialization equations are derived from the polar form of a weight $w$ of $W$:

$$w = |w|e^{q_{imag}^{\triangleleft}\theta} = |w|(cos(\theta) + q_{imag}^{\triangleleft}sin(\theta)), \tag{14}$$

and,

$$w_{\mathbf{r}} = \varphi\, cos(\theta), \quad w_{\mathbf{i}} = \varphi\, q_{imag\mathbf{i}}^{\triangleleft}\, sin(\theta), \quad w_{\mathbf{j}} = \varphi\, q_{imag\mathbf{j}}^{\triangleleft}\, sin(\theta), \quad w_{\mathbf{k}} = \varphi\, q_{imag\mathbf{k}}^{\triangleleft}\, sin(\theta). \tag{15}$$

The angle $\theta$ is randomly generated in the interval $[-\pi, \pi]$. The quaternion $q_{imag}^{\triangleleft}$ is defined as purely normalized imaginary, and is expressed as $q_{imag}^{\triangleleft} = 0 + x\mathbf{i} + y\mathbf{j} + z\mathbf{k}$. The imaginary components $x\mathbf{i}$, $y\mathbf{j}$, and $z\mathbf{k}$ are sampled from an uniform distribution in $[0, 1]$ to obtain $q_{imag}$, which is then normalized (following Eq. 4) to obtain $q_{imag}^{\triangleleft}$. The parameter $\varphi$ is a random number generated with respect to well-known initialization criterions (such as Glorot or He algorithms) (Glorot & Bengio, 2010; He et al., 2015). However, the equations derived in (Glorot & Bengio, 2010; He et al., 2015) are defined for real-valued weight matrices. Therefore, the variance of $W$ has to be investigated in the quaternion space to obtain $\varphi$ (the full demonstration is provided in Appendix 6.2). The variance of $W$ is:

$$Var(W) = \mathbb{E}(|W|^2) - [\mathbb{E}(|W|)]^2, \text{ with } [\mathbb{E}(|W|)]^2 = 0. \tag{16}$$

---

**Algorithm 1** Quaternion-valued weight initialization

---

1: **procedure** QINIT($W, n_{in}, n_{out}$)
2: $\quad \sigma \leftarrow \frac{1}{\sqrt{2(n_{in}+n_{out})}}$ $\qquad\qquad\qquad\qquad\qquad$ ▷ *w.r.t to Glorot criterion and Eq. 18*
3: $\quad$ **for** $w$ in $W$ **do**
4: $\qquad \theta \leftarrow rand(-\pi, \pi)$
5: $\qquad \varphi \leftarrow rand(-\sigma, \sigma)$
6: $\qquad x, y, z \leftarrow rand(0, 1)$
7: $\qquad q_{imag} \leftarrow Quaternion(0, x, y, z)$
8: $\qquad q_{imag}^{\lhd} \leftarrow \frac{q_{imag}}{\sqrt{x^2+y^2+z^2}}$
9: $\qquad w_r \leftarrow \varphi \times cos(\theta)$ $\qquad\qquad\qquad\qquad\qquad\qquad$ ▷ *See Eq. 15*
10: $\qquad w_i \leftarrow \varphi \times q_{imag_i}^{\lhd} \times sin(\theta)$
11: $\qquad w_j \leftarrow \varphi \times q_{imag_j}^{\lhd} \times sin(\theta)$
12: $\qquad w_k \leftarrow \varphi \times q_{imag_k}^{\lhd} \times sin(\theta)$
13: $\qquad w \leftarrow Quaternion(w_r, w_i, w_j, w_k)$

---

Indeed, the weight distribution is normalized. The value of $Var(W) = \mathbb{E}(|W|^2)$, instead, is not trivial in the case of quaternion-valued matrices. Indeed, $W$ follows a Chi-distribution with four degrees of freedom (DOFs). Consequently, $Var(W)$ is expressed and computed as follows:

$$Var(W) = \mathbb{E}(|W|^2) = \int_0^\infty x^2 f(x)\,\mathrm{d}x = 4\sigma^2. \tag{17}$$

The Glorot (Glorot & Bengio, 2010) and He (He et al., 2015) criterions are extended to quaternion as:

$$\sigma = \frac{1}{\sqrt{2(n_{in}+n_{out})}}, \text{ and } \sigma = \frac{1}{\sqrt{2n_{in}}}, \tag{18}$$

with $n_{in}$ and $n_{out}$ the number of neurons of the input and output layers respectively. Finally, $\varphi$ can be sampled from $[-\sigma, \sigma]$ to complete the weight initialization of Eq. 15.

## 4 EXPERIMENTS

This Section details the acoustic features extraction (Section 4.1), the experimental setups and the results obtained with QRNNs, QLSTMs, RNNs and LSTMs on the TIMIT speech recognition tasks (Section 4.2). The results reported in bold on tables are obtained with the best configurations of the neural networks observed with the validation set.

### 4.1 QUATERNION ACOUSTIC FEATURES

The raw audio is first splitted every 10ms with a window of 25ms. Then 40-dimensional log Mel-filter-bank coefficients with first, second, and third order derivatives are extracted using the *pytorch-kaldi*[2] (Ravanelli et al., 2018b) toolkit and the Kaldi s5 recipes (Povey et al., 2011). An acoustic quaternion $Q(f, t)$ associated with a frequency $f$ and a time-frame $t$ is formed as follows:

$$Q(f, t) = e(f, t) + \frac{\partial e(f, t)}{\partial t}\mathbf{i} + \frac{\partial^2 e(f, t)}{\partial^2 t}\mathbf{j} + \frac{\partial^3 e(f, t)}{\partial^3 t}\mathbf{k}. \tag{19}$$

$Q(f, t)$ represents multiple views of a frequency $f$ at time frame $t$, consisting of the energy $e(f, t)$ in the filter band at frequency $f$, its first time derivative describing a slope view, its second time derivative describing a concavity view, and the third derivative describing the rate of change of the second derivative. Quaternions are used to learn the spatial relations that exist between the 3 described different views that characterize a same frequency (Tokuda et al., 2003). Thus, the quaternion input vector length is $160/4 = 40$. Decoding is based on Kaldi (Povey et al., 2011) and weighted finite state transducers (WFST) (Mohri et al., 2002) that integrate acoustic, lexicon and language model probabilities into a single HMM-based search graph.

---

[2]pytorch-kaldi is available at `https://github.com/mravanelli/pytorch-kaldi`

## 4.2 THE TIMIT CORPUS

The training process is based on the standard $3,696$ sentences uttered by $462$ speakers, while testing is conducted on $192$ sentences uttered by $24$ speakers of the TIMIT (Garofolo et al., 1993) dataset. A validation set composed of $400$ sentences uttered by $50$ speakers is used for hyper-parameter tuning. The models are compared on a fixed number of layers $M = 4$ and by varying the number of neurons $N$ from 256 to $2,048$, and 64 to $512$ for the RNN and QRNN respectively. Indeed, it is worth underlying that the number of hidden neurons in the quaternion and real spaces do not handle the same amount of real-number values. Indeed, 256 quaternion neurons output are $256 \times 4 = 1024$ real values. Tanh activations are used across all the layers except for the output layer that is based on a softmax function. Models are optimized with RMSPROP with vanilla hyper-parameters and an initial learning rate of $8 \cdot 10^{-4}$. The learning rate is progressively annealed using a halving factor of $0.5$ that is applied when no performance improvement on the validation set is observed. The models are trained during 25 epochs. All the models converged to a minimum loss, due to the annealed learning rate. A dropout rate of $0.2$ is applied over all the hidden layers (Srivastava et al., 2014) except the output one. The negative log-likelihood loss function is used as an objective function. All the experiments are repeated 5 times (5-folds) with different seeds and are averaged to limit any variation due to the random initialization.

Table 1: Phoneme error rate (PER%) of QRNN and RNN models on the development and test sets of the TIMIT dataset. "Params" stands for the total number of trainable parameters.

| Models | Neurons | Dev. | Test | Params |
|--------|---------|------|------|--------|
|        | 256     | 22.4 | 23.4 | 1M     |
| RNN    | 512     | 19.6 | 20.4 | 2.8M   |
|        | **1,024** | **17.9** | **19.0** | **9.4M** |
|        | 2,048   | 20.0 | 20.7 | 33.4M  |
|        | 64      | 23.6 | 23.9 | 0.6M   |
| QRNN   | 128     | 19.2 | 20.1 | 1.4M   |
|        | **256** | **17.4** | **18.5** | **3.8M** |
|        | 512     | 17.5 | 18.7 | 11.2M  |

The results on the TIMIT task are reported in Table 1. The best PER in realistic conditions (w.r.t to the best validation PER) is $18.5\%$ and $19.0\%$ on the test set for QRNN and RNN models respectively, highlighting an absolute improvement of $0.5\%$ obtained with QRNN. These results compare favorably with the best results obtained so far with architectures that do not integrate access control in multiple memory layers (Ravanelli et al., 2018a). In the latter, a PER of $18.3\%$ is reported on the TIMIT test set with batch-normalized RNNs . Moreover, a remarkable advantage of QRNNs is a drastic reduction (with a factor of $2.5\times$) of the parameters needed to achieve these results. Indeed, such PERs are obtained with models that employ the same internal dimensionality corresponding to $1,024$ real-valued neurons and 256 quaternion-valued ones, resulting in a number of parameters of 3.8M for QRNN against the 9.4M used in the real-valued RNN. It is also worth noting that QRNNs consistently need fewer parameters than equivalently sized RNNs, with an average reduction factor of 2.26 times. This is easily explained by considering the content of the quaternion algebra. Indeed, for a fully-connected layer with $2,048$ input values and $2,048$ hidden units, a real-valued RNN has $2,048^2 \approx 4.2M$ parameters, while to maintain equal input and output dimensions the quaternion equivalent has 512 quaternions inputs and 512 quaternion hidden units. Therefore, the number of parameters for the quaternion-valued model is $512^2 \times 4 \approx 1M$. Such a complexity reduction turns out to produce better results and has other advantages such as a smaller memory footprint while saving models on budget memory systems. This characteristic makes our QRNN model particularly suitable for speech recognition conducted on low computational power devices like smartphones (Chen et al., 2014). QRNNs and RNNs accuracies vary accordingly to the architecture with better PER on bigger and wider topologies. Therefore, while good PER are observed with a higher number of parameters, smaller architectures performed at $23.9\%$ and $23.4\%$, with 1M and 0.6M parameters for the RNN and the QRNN respectively. Such PER are due to a too small number of parameters to solve the task.

## 4.3 QUATERNION LONG-SHORT TERM MEMORY NEURAL NETWORKS

We propose to extend the QRNN to state-of-the-art models such as long-short term memory neural networks (LSTM), to support and improve the results already observed with the QRNN compared to the RNN in more realistic conditions. LSTM (Hochreiter & Schmidhuber, 1997) neural networks

were introduced to solve the problems of long-term dependencies learning and vanishing or exploding gradient observed with long sequences. Based on the equations of the forward propagation and back propagation through time of QRNN described in Section 3.3.1, and Section 3.3.2, one can easily derive the equations of a quaternion-valued LSTM. Gates are defined with quaternion numbers following the proposal of Danihelka et al. (2016). Therefore, the gate action is characterized by an independent modification of each component of the quaternion-valued signal following a component-wise product with the quaternion-valued gate potential. Let $f_t$, $i_t$, $o_t$, $c_t$, and $h_t$ be the forget, input, output gates, cell states and the hidden state of a LSTM cell at time-step $t$:

$$f_t = \alpha(W_f \otimes x_t + R_f \otimes h_{t-1} + b_f), \tag{20}$$
$$i_t = \alpha(W_i \otimes x_t + R_i \otimes h_{t-1} + b_i), \tag{21}$$
$$c_t = f_t \times c_{t-1} + i_t \times tanh(W_c \otimes x_t + R_c \otimes h_{t-1} + b_c), \tag{22}$$
$$o_t = \alpha(W_o \otimes x_t + R_o \otimes h_{t-1} + b_o), \tag{23}$$
$$h_t = o_t \times tanh(c_t), \tag{24}$$

where $W$ are rectangular input weight matrices, $R$ are square recurrent weight matrices, and $b$ are bias vectors. $\alpha$ is the split activation function and $\times$ denotes a component-wise product between two quaternions. Both QLSTM and LSTM are bidirectional and trained on the same conditions than for the QRNN and RNN experiments.

Table 2: Phoneme error rate (PER%) of QLSTM and LSTM models on the development and test sets of the TIMIT dataset. "Params" stands for the total number of trainable parameters.

| Models | Neurons | Dev. | Test | Params |
|--------|---------|------|------|--------|
| LSTM   | 256     | 14.9 | 16.5 | 3.6M   |
|        | 512     | 14.2 | 16.1 | 12.6M  |
|        | **1,024** | **14.4** | **15.3** | **46.2M** |
|        | 2,048   | 14.0 | 15.9 | 176.3M |
| QLSTM  | 64      | 15.5 | 17.0 | 1.6M   |
|        | 128     | 14.1 | 16.0 | 4.6M   |
|        | **256** | **14.0** | **15.1** | **14.4M** |
|        | 512     | 14.2 | 15.1 | 49.9M  |

The results on the TIMIT corpus reported on Table 2 support the initial intuitions and the previously established trends. We first point out that the best PER observed is $15.1\%$ and $15.3\%$ on the test set for QLSTMs and LSTM models respectively with an absolute improvement of $0.2\%$ obtained with QLSTM using 3.3 times fewer parameters compared to LSTM. These results are among the top of the line results (Graves et al., 2013b; Ravanelli et al., 2018a) and prove that the proposed quaternion approach can be used in state-of-the-art models. A deeper investigation of QLSTMs performances with the larger Wall Street Journal (WSJ) dataset can be found in Appendix 6.1.1.

## 5   CONCLUSION

**Summary**. This paper proposes to process sequences of multidimensional features (such as acoustic data) with a novel quaternion recurrent neural network (QRNN) and quaternion long-short term memory neural network (QLSTM). The experiments conducted on the TIMIT phoneme recognition task show that QRNNs and QLSTMs are more effective to learn a compact representation of multidimensional information by outperforming RNNs and LSTMs with 2 to 3 times less free parameters. Therefore, our initial intuition that the quaternion algebra offers a better and more compact representation for multidimensional features, alongside with a better learning capability of feature internal dependencies through the *Hamilton product*, have been demonstrated.

**Future Work**. Future investigations will develop other multi-view features that contribute to decrease ambiguities in representing phonemes in the quaternion space. In this extent, a recent approach based on a quaternion Fourier transform to create quaternion-valued signal has to be investigated. Finally, other high-dimensional neural networks such as manifold and Clifford networks remain mostly unexplored and can benefit from further research.

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

## 6 APPENDIX

### 6.1 WALL STREET JOURNAL EXPERIMENTS AND COMPUTATIONAL COMPLEXITY

This Section proposes to validate the scaling of the proposed QLSTMs to a bigger and more realistic corpus, with a speech recognition task on the Wall Street Journal (WSJ) dataset. Finally, it discuses the impact of the quaternion algebra in term of computational compexity.

#### 6.1.1 SPEECH RECOGNITION WITH THE WALL STREET JOURNAL CORPUS

We propose to evaluate both QLSTMs and LSTMs with a larger and more realistic corpus to validate the scaling of the observed TIMIT results (Section 4.2). Acoustic input features are described in Section 4.1, and extracted on both the $14$ hour subset 'train-si84', and the full $81$ hour dataset 'train-si284' of the Wall Street Journal (WSJ) corpus. The 'test-dev93' development set is employed for validation, while 'test-eval92' composes the testing set. Models architectures are fixed with respect to the best results observed with the TIMIT corpus (Section 4.2). Therefore, both QLSTMs and LSTMs contain four bidirectional layers of internal dimension of size $1,024$. Then, an additional layer of internal size $1,024$ is added before the output layer. The only change on the training procedure compared to the TIMIT experiments concerns the model optimizer, which is set to Adam (Kingma & Ba, 2014) instead of RMSPROP. Results are from a 3-folds average.

Table 3: Word error rates (WER %) obtained with both training set (WSJ14h and WSJ81h) of the Wall Street Journal corpus. 'test-dev93' and 'test-eval92' are used as validation and testing set respectively. $L$ expresses the number of recurrent layers.

| Models | WSJ14 Dev. | WSJ14 Test | WSJ81 Dev. | WSJ81 Test | Params |
|--------|-----------|-----------|-----------|-----------|--------|
| LSTM | 11.2 | 7.2 | 7.4 | 4.5 | 53.7M |
| QLSTM | 10.9 | 6.9 | 7.2 | 4.3 | 18.7M |

It is important to notice that reported results on Table 3 compare favorably with equivalent architectures (Graves et al., 2013a) (WER of $11.7\%$ on 'test-dev93'), and are competitive with state-of-the-art and much more complex models based on better engineered features (Chan & Lane, 2015)(WER of $3.8\%$ with the 81 hours of training data, and on 'test-eval92'). According to Table 3, QLSTMs outperform LSTM in all the training conditions ($14$ hours and $81$ hours) and with respect to both the validation and testing sets. Moreover, QLSTMs still need 2.9 times less neural parameters than LSTMs to achieve such performances. This experiment demonstrates that QLSTMs scale well to larger and more realistic speech datasets and are still more efficient than real-valued LSTMs.

#### 6.1.2 NOTES ON COMPUTATIONAL COMPLEXITY

A computational complexity of $O(n^2)$ with $n$ the number of hidden states has been reported by Morchid (2018) for real-valued LSTMs. QLSTMs just involve $4$ times larger matrices during computations. Therefore, the computational complexity remains unchanged and equals to $O(n^2)$. Nonetheless, and due to the *Hamilton product*, a single forward propagation between two quaternion neurons uses $28$ operations, compared to a single one for two real-valued neurons, implying a longer training time (up to $3$ times slower). However, such worst speed performances could easily be alleviated with a proper engineered cuDNN kernel for the *Hamilton product*, that would helps QNNs to be more efficient than real-valued ones. A well-adapted CUDA kernel would allow QNNs to perform more computations, with fewer parameters, and therefore less memory copy operations from the CPU to the GPU.

### 6.2 PARAMETERS INITIALIZATION

Let us recall that a generated quaternion weight $w$ from a weight matrix $W$ has a polar form defined as:

$$w = |w|e^{q_{imag}^{\triangleleft}\theta} = |w|(cos(\theta) + q_{imag}^{\triangleleft}sin(\theta)), \tag{25}$$

with $q^{\triangleleft}_{imag} = 0 + x\mathbf{i} + y\mathbf{j} + z\mathbf{k}$ a purely imaginary and normalized quaternion. Therefore, $w$ can be computed following:

$$
\begin{aligned}
w_{\mathbf{r}} &= \varphi \, cos(\theta), \\
w_{\mathbf{i}} &= \varphi \, q^{\triangleleft}_{imag\mathbf{i}} \, sin(\theta), \\
w_{\mathbf{j}} &= \varphi \, q^{\triangleleft}_{imag\mathbf{j}} \, sin(\theta), \\
w_{\mathbf{k}} &= \varphi \, q^{\triangleleft}_{imag\mathbf{k}} \, sin(\theta).
\end{aligned}
\tag{26}
$$

However, $\varphi$ represents a randomly generated variable with respect to the variance of the quaternion weight and the selected initialization criterion. The initialization process follows (Glorot & Bengio, 2010) and (He et al., 2015) to derive the variance of the quaternion-valued weight parameters. Indeed, the variance of $\mathbf{W}$ has to be investigated:

$$
Var(W) = \mathbb{E}(|W|^2) - [\mathbb{E}(|W|)]^2.
\tag{27}
$$

$[\mathbb{E}(|W|)]^2$ is equals to $0$ since the weight distribution is symmetric around $0$. Nonetheless, the value of $Var(W) = \mathbb{E}(|W|^2)$ is not trivial in the case of quaternion-valued matrices. Indeed, $W$ follows a Chi-distribution with four degrees of freedom (DOFs) and $\mathbb{E}(|W|^2)$ is expressed and computed as follows:

$$
\mathbb{E}(|W|^2) = \int_0^\infty x^2 f(x) \, \mathrm{d}x,
\tag{28}
$$

With $f(x)$ is the probability density function with four DOFs. A four-dimensional vector $X = \{A, B, C, D\}$ is considered to evaluate the density function $f(x)$. $X$ has components that are normally distributed, centered at zero, and independent. Then, $A$, $B$, $C$ and $D$ have density functions:

$$
f_A(x; \sigma) = f_B(x; \sigma) = f_C(x; \sigma) = f_D(x; \sigma) = \frac{e^{-x^2/2\sigma^2}}{\sqrt{2\pi\sigma^2}}.
\tag{29}
$$

The four-dimensional vector $X$ has a length $L$ defined as $L = \sqrt{A^2 + B^2 + C^2 + D^2}$ with a cumulative distribution function $F_L(x; \sigma)$ in the 4-sphere (n-sphere with $n = 4$) $S_x$:

$$
F_L(x; \sigma) = \int \int \int \int_{S_x} f_A(x; \sigma) f_B(x; \sigma) f_C(x; \sigma) f_D(x; \sigma) \, \mathrm{d}S_x
\tag{30}
$$

where $S_x = \{(a, b, c, d) : \sqrt{a^2 + b^2 + c^2 + d^2} < x\}$ and $\mathrm{d}S_x = \mathrm{d}a \, \mathrm{d}b \, \mathrm{d}c \, \mathrm{d}d$. The polar representations of the coordinates of $X$ in a 4-dimensional space are defined to compute $\mathrm{d}S_x$:

$$
\begin{aligned}
a &= \rho \cos\theta, \\
b &= \rho \sin\theta \cos\phi, \\
c &= \rho \sin\theta \sin\phi \cos\psi, \\
d &= \rho \sin\theta \sin\phi \sin\psi,
\end{aligned}
$$

where $\rho$ is the magnitude ($\rho = \sqrt{a^2 + b^2 + c^2 + d^2}$) and $\theta$, $\phi$, and $\psi$ are the phases with $0 \le \theta \le \pi$, $0 \le \phi \le \pi$ and $0 \le \psi \le 2\pi$. Then, $\mathrm{d}S_x$ is evaluated with the Jacobian $J_f$ of $f$ defined as:

$$
J_f = \frac{\partial(a, b, c, d)}{\partial(\rho, \theta, \phi, \psi)} = \frac{\mathrm{d}a \, \mathrm{d}b \, \mathrm{d}c \, \mathrm{d}d}{\mathrm{d}\rho \, \mathrm{d}\theta \, \mathrm{d}\phi \, \mathrm{d}\psi} = \begin{vmatrix} \frac{\mathrm{d}a}{\mathrm{d}\rho} & \frac{\mathrm{d}a}{\mathrm{d}\theta} & \frac{\mathrm{d}a}{\mathrm{d}\phi} & \frac{\mathrm{d}a}{\mathrm{d}\psi} \\ \frac{\mathrm{d}b}{\mathrm{d}\rho} & \frac{\mathrm{d}b}{\mathrm{d}\theta} & \frac{\mathrm{d}b}{\mathrm{d}\phi} & \frac{\mathrm{d}b}{\mathrm{d}\psi} \\ \frac{\mathrm{d}c}{\mathrm{d}\rho} & \frac{\mathrm{d}c}{\mathrm{d}\theta} & \frac{\mathrm{d}c}{\mathrm{d}\phi} & \frac{\mathrm{d}c}{\mathrm{d}\psi} \\ \frac{\mathrm{d}d}{\mathrm{d}\rho} & \frac{\mathrm{d}d}{\mathrm{d}\theta} & \frac{\mathrm{d}d}{\mathrm{d}\phi} & \frac{\mathrm{d}d}{\mathrm{d}\psi} \end{vmatrix}
$$

$$
= \begin{vmatrix} \cos\theta & -\rho\sin\theta & 0 & 0 \\ \sin\theta\cos\phi & \rho\sin\theta\cos\phi & -\rho\sin\theta\sin\phi & 0 \\ \sin\theta\sin\phi\cos\psi & \rho\cos\theta\sin\phi\cos\psi & \rho\sin\theta\cos\phi\cos\psi & -\rho\sin\theta\sin\phi\sin\psi \\ \sin\theta\sin\phi\sin\psi & \rho\cos\theta\sin\phi\sin\psi & \rho\sin\theta\cos\phi\sin\psi & \rho\sin\theta\sin\phi\cos\psi \end{vmatrix}.
$$

And,

$$
J_f = \rho^3 \sin^2\theta \sin\phi.
\tag{31}
$$

Therefore, by the Jacobian $J_f$, we have the polar form:

$$\mathrm{d}a\,\mathrm{d}b\,\mathrm{d}c\,\mathrm{d}d = \rho^3 \sin^2\theta \sin\phi\,\mathrm{d}\rho\,\mathrm{d}\theta\,\mathrm{d}\phi\,\mathrm{d}\psi. \tag{32}$$

Then, writing Eq.(30) in polar coordinates, we obtain:

$$\begin{aligned}
F_L(x,\sigma) &= \left(\frac{1}{\sqrt{2\pi\sigma^2}}\right)^4 \int\int\int\int_0^x e^{-a^2/2\sigma^2} e^{-b^2/2\sigma^2} e^{-c^2/2\sigma^2} e^{-d^2/2\sigma^2}\,\mathrm{d}S_x \\
&= \frac{1}{4\pi^2\sigma^4} \int_0^{2\pi}\int_0^{\pi}\int_0^{\pi}\int_0^x e^{-\rho^2/2\sigma^2} \rho^3 \sin^2\theta \sin\phi\,\mathrm{d}\rho\,\mathrm{d}\theta\,\mathrm{d}\phi\,\mathrm{d}\psi \\
&= \frac{1}{4\pi^2\sigma^4} \int_0^{2\pi}\mathrm{d}\psi \int_0^{\pi}\sin\phi\,\mathrm{d}\phi \int_0^{\pi}\sin^2\theta\,\mathrm{d}\theta \int_0^x \rho^3 e^{-\rho^2/2\sigma^2}\,\mathrm{d}\rho \\
&= \frac{1}{4\pi^2\sigma^4} 2\pi 2 \left[\frac{\theta}{2} - \frac{\sin 2\theta}{4}\right]_0^{\pi} \int_0^x \rho^3 e^{-\rho^2/2\sigma^2}\,\mathrm{d}\rho \\
&= \frac{1}{4\pi^2\sigma^4} 4\pi\frac{\pi}{2} \int_0^x \rho^3 e^{-\rho^2/2\sigma^2}\,\mathrm{d}\rho,
\end{aligned}$$

Then,

$$F_L(x,\sigma) = \frac{1}{2\sigma^4} \int_0^x \rho^3 e^{-\rho^2/2\sigma^2}\,\mathrm{d}\rho. \tag{33}$$

The probability density function for $X$ is the derivative of its cumulative distribution function, which by the fundamental theorem of calculus is:

$$\begin{aligned}
f_L(x,\sigma) &= \frac{\mathrm{d}}{\mathrm{d}x} F_L(x,\sigma) \\
&= \frac{1}{2\sigma^4} x^3 e^{-x^2/2\sigma^2}. \tag{34}
\end{aligned}$$

The expectation of the squared magnitude becomes:

$$\begin{aligned}
\mathbb{E}(|W|^2) &= \int_0^{\infty} x^2 f(x)\,\mathrm{d}x \\
&= \int_0^{\infty} x^2 \frac{1}{2\sigma^4} x^3 e^{-x^2/2\sigma^2}\,\mathrm{d}x \\
&= \frac{1}{2\sigma^4} \int_0^{\infty} x^5 e^{-x^2/2\sigma^2}\,\mathrm{d}x.
\end{aligned}$$

With integration by parts we obtain:

$$\begin{aligned}
\mathbb{E}(|W|^2) &= \frac{1}{2\sigma^4} \left(-x^4\sigma^2 e^{-x^2/2\sigma^2}\Big|_0^{\infty} + \int_0^{\infty} \sigma^2 4x^3 e^{-x^2/2\sigma^2}\,\mathrm{d}x\right) \\
&= \frac{1}{2\sigma^2} \left(-x^4 e^{-x^2/2\sigma^2}\Big|_0^{\infty} + \int_0^{\infty} 4x^3 e^{-x^2/2\sigma^2}\,\mathrm{d}x\right). \tag{35}
\end{aligned}$$

The expectation $\mathbb{E}(|W|^2)$ is the sum of two terms. The first one:

$$\begin{aligned}
-x^4 e^{-x^2/2\sigma^2}\Big|_0^{\infty} &= \lim_{x\to+\infty} -x^4 e^{-x^2/2\sigma^2} - \lim_{x\to+0} x^4 e^{-x^2/2\sigma^2} \\
&= \lim_{x\to+\infty} -x^4 e^{-x^2/2\sigma^2},
\end{aligned}$$

Based on the L'Hôpital's rule, the undetermined limit becomes:

$$
\begin{aligned}
\lim_{x \to +\infty} -x^4 e^{-x^2/2\sigma^2} &= -\lim_{x \to +\infty} \frac{x^4}{e^{x^2/2\sigma^2}} \\
&= \dots \\
&= -\lim_{x \to +\infty} \frac{24}{(1/\sigma^2)(P(x)e^{x^2/2\sigma^2})} \\
&= 0.
\end{aligned}
\tag{36}
$$

With $P(x)$ is polynomial and has a limit to $+\infty$. The second term is calculated in a same way (integration by parts) and $\mathbb{E}(|W|^2)$ becomes from Eq.(35):

$$
\begin{aligned}
\mathbb{E}(|W|^2) &= \frac{1}{2\sigma^2} \int_0^\infty 4x^3 e^{-x^2/2\sigma^2} \, \mathrm{d}x \\
&= \frac{2}{\sigma^2} \left( x^2 \sigma^2 e^{-x^2/2\sigma^2} \Big|_0^\infty + \int_0^\infty \sigma^2 2x e^{-x^2/2\sigma^2} \, \mathrm{d}x \right).
\end{aligned}
\tag{37}
$$

The limit of first term is equals to 0 with the same method than in Eq.(36). Therefore, the expectation is:

$$
\begin{aligned}
\mathbb{E}(|W|^2) &= 4 \left( \int_0^\infty x e^{-x^2/2\sigma^2} \, \mathrm{d}x \right) \\
&= 4\sigma^2.
\end{aligned}
\tag{38}
$$

And finally the variance is:

$$
Var(|W|) = 4\sigma^2.
\tag{39}
$$

### 6.3 QUATERNION BACKPROPAGATION THROUGH TIME

Let us recall the forward equations and parameters needed to derive the complete quaternion backpropagation through time (QBPTT) algorithm.

#### 6.3.1 RECALL OF THE FORWARD PHASE

Let $x_t$ be the input vector at timestep $t$, $h_t$ the hidden state, $W_{hh}$, $W_{xh}$ and $W_{hy}$ the hidden state, input and output weight matrices respectively. Finally $b_h$ is the biases vector of the hidden states and $p_t$, $y_t$ are the output and the expected target vector.

$$
h_t = \alpha(h_t^{preact}),
\tag{40}
$$

with,

$$
h_t^{preact} = W_{hh} \otimes h_{t-1} + W_{xh} \otimes x_t + b_h,
\tag{41}
$$

and $\alpha$ is the quaternion split activation function (Xu et al., 2017) of a quaternion $Q$ defined as:

$$
\alpha(Q) = f(r) + if(x) + jf(y) + kf(z),
\tag{42}
$$

and $f$ corresponding to any standard activation function. The output vector $p_t$ can be computed as:

$$
p_t = \beta(p_t^{preact}),
\tag{43}
$$

with

$$
p_t^{preact} = W_{hy} \otimes h_t,
\tag{44}
$$

and $\beta$ any split activation function. Finally, the objective function is a real-valued loss function applied component-wise. The gradient with respect to the MSE loss is expressed for each weight matrix as $\frac{\partial E_t}{\partial W_{hy}}$, $\frac{\partial E_t}{\partial W_{hh}}$, $\frac{\partial E_t}{\partial W_{hx}}$, and for the bias vector as $\frac{\partial E_t}{\partial B_h}$. In the real-valued space, the dynamic of the loss is only investigated based on all previously connected neurons. In this extent, the QBPTT differs from BPTT due to the fact that the loss must also be derived with respect to each component of a quaternion neural parameter, making it bi-level. This could act as a regularizer during the training process.

### 6.3.2 OUTPUT WEIGHT MATRIX

The weight matrix $W_{hy}$ is used only in the computation of $p_t$. It is therefore straightforward to compute $\frac{\partial E_t}{\partial W_{hy}}$:

$$\frac{\partial E_t}{\partial W_{hy}} = \frac{\partial E_t}{\partial W_{hy}^r} + i\frac{\partial E_t}{\partial W_{hy}^i} + j\frac{\partial E_t}{\partial W_{hy}^j} + k\frac{\partial E_t}{\partial W_{hy}^k}. \tag{45}$$

Each quaternion component is then derived following the chain rule:

$$\begin{aligned}
\frac{\partial E_t}{\partial W_{hy}^r} &= \frac{\partial E_t}{\partial p_t^r}\frac{\partial p_t^r}{\partial W_{hy}^r} + \frac{\partial E_t}{\partial p_t^i}\frac{\partial p_t^i}{\partial W_{hy}^r} + \frac{\partial E_t}{\partial p_t^j}\frac{\partial p_t^j}{\partial W_{hy}^r} + \frac{\partial E_t}{\partial p_t^k}\frac{\partial p_t^k}{\partial W_{hy}^r} \\
&= (p_t^r - y_t^r) \times h_t^r + (p_t^i - y_t^i) \times h_t^i + (p_t^j - y_t^j) \times h_t^j + (p_t^k - y_t^k) \times h_t^k.
\end{aligned} \tag{46}$$

$$\begin{aligned}
\frac{\partial E_t}{\partial W_{hy}^i} &= \frac{\partial E_t}{\partial p_t^r}\frac{\partial p_t^r}{\partial W_{hy}^i} + \frac{\partial E_t}{\partial p_t^i}\frac{\partial p_t^i}{\partial W_{hy}^i} + \frac{\partial E_t}{\partial p_t^j}\frac{\partial p_t^j}{\partial W_{hy}^i} + \frac{\partial E_t}{\partial p_t^k}\frac{\partial p_t^k}{\partial W_{hy}^i} \\
&= (p_t^r - y_t^r) \times -h_t^i + (p_t^i - y_t^i) \times h_t^r + (p_t^j - y_t^j) \times h_t^k + (p_t^k - y_t^k) \times -h_t^j.
\end{aligned} \tag{47}$$

$$\begin{aligned}
\frac{\partial E_t}{\partial W_{hy}^j} &= \frac{\partial E_t}{\partial p_t^r}\frac{\partial p_t^r}{\partial W_{hy}^j} + \frac{\partial E_t}{\partial p_t^i}\frac{\partial p_t^i}{\partial W_{hy}^j} + \frac{\partial E_t}{\partial p_t^j}\frac{\partial p_t^j}{\partial W_{hy}^j} + \frac{\partial E_t}{\partial p_t^k}\frac{\partial p_t^k}{\partial W_{hy}^j} \\
&= (p_t^r - y_t^r) \times -h_t^j + (p_t^i - y_t^i) \times -h_t^k + (p_t^j - y_t^j) \times h_t^r + (p_t^k - y_t^k) \times h_t^i.
\end{aligned} \tag{48}$$

$$\begin{aligned}
\frac{\partial E_t}{\partial W_{hy}^k} &= \frac{\partial E_t}{\partial p_t^r}\frac{\partial p_t^r}{\partial W_{hy}^k} + \frac{\partial E_t}{\partial p_t^i}\frac{\partial p_t^i}{\partial W_{hy}^k} + \frac{\partial E_t}{\partial p_t^j}\frac{\partial p_t^j}{\partial W_{hy}^k} + \frac{\partial E_t}{\partial p_t^k}\frac{\partial p_t^k}{\partial W_{hy}^k} \\
&= (p_t^r - y_t^r) \times -h_t^k + (p_t^i - y_t^i) \times h_t^j + (p_t^j - y_t^j) \times -h_t^i + (p_t^k - y_t^k) \times h_t^r.
\end{aligned} \tag{49}$$

By regrouping in a matrix form the $h_t$ components from these equations, one can define:

$$\begin{bmatrix} h_t^r & h_t^i & h_t^j & h_t^k \\ -h_t^i & h_t^r & h_t^k & -h_t^j \\ -h_t^j & -h_t^k & h_t^r & h_t^i \\ -h_t^k & h_t^j & -h_t^i & h_t^r \end{bmatrix} = h_t^*. \tag{50}$$

Therefore,

$$\frac{\partial E_t}{\partial W_{hy}} = (p_t - y_t) \otimes h_t^*. \tag{51}$$

### 6.3.3 HIDDEN WEIGHT MATRIX

Conversely to $W_{hy}$ the weight matrix $W_{hh}$ is an argument of $h_t$ with $h_{t-1}$ involved. The recursive backpropagation can thus be derived as:

$$\frac{\partial E}{\partial W_{hh}} = \sum_{t=0}^{N} \frac{\partial E_t}{\partial W_{hh}}. \tag{52}$$

And,

$$\frac{\partial E_t}{\partial W_{hh}} = \sum_{m=0}^{t} \frac{\partial E_m}{\partial W_{hh}^r} + i\frac{\partial E_m}{\partial W_{hh}^r} + j\frac{\partial E_m}{\partial W_{hh}^i} + k\frac{\partial E_m}{\partial W_{hh}^k}, \tag{53}$$

with $N$ the number of timesteps that compose the sequence. As for $W_{hy}$ we start with $\frac{\partial E_k}{\partial W_{hh}^r}$:

$$\begin{aligned}
\sum_{m=0}^{t} \frac{\partial E_m}{\partial W_{hh}^r} &= \sum_{m=0}^{t} \frac{\partial E_t}{\partial h_t^r}\frac{\partial h_t^r}{\partial h_m^r}\frac{\partial h_m^r}{\partial W_{hh}^r} + \frac{\partial E_t}{\partial h_t^i}\frac{\partial h_t^i}{\partial h_m^i}\frac{\partial h_m^i}{\partial W_{hh}^r} \\
&\quad + \frac{\partial E_t}{\partial h_t^j}\frac{\partial h_t^j}{\partial h_m^j}\frac{\partial h_m^j}{\partial W_{hh}^r} + \frac{\partial E_t}{\partial h_t^k}\frac{\partial h_t^k}{\partial h_m^k}\frac{\partial h_m^k}{\partial W_{hh}^r}.
\end{aligned} \tag{54}$$

Non-recursive elements are derived w.r.t r, **i,j**, **k**:

$$
\begin{aligned}
\frac{\partial E_t}{\partial h_t^r} &= \frac{\partial E_t}{\partial p_t^r}\frac{\partial p_t^r}{\partial h_t^r} + \frac{\partial E_t}{\partial p_t^i}\frac{\partial p_t^i}{\partial h_t^r} + \frac{\partial E_t}{\partial p_t^j}\frac{\partial p_t^j}{\partial h_t^r} + \frac{\partial E_t}{\partial p_t^k}\frac{\partial p_t^k}{\partial h_t^r} \\
&= (p_t^r - y_t^r) \times f'(p_t^r) \times W_{hy}^r + (p_t^i - y_t^i) \times f'(p_t^i) \times W_{hy}^i \\
&\quad + (p_t^j - y_t^j) \times f'(p_t^j) \times W_{hy}^j + (p_t^k - y_t^k) \times f'(p_t^k) \times W_{hy}^k.
\end{aligned}
\tag{55}
$$

$$
\begin{aligned}
\frac{\partial E_t}{\partial h_t^i} &= \frac{\partial E_t}{\partial p_t^r}\frac{\partial p_t^r}{\partial h_t^i} + \frac{\partial E_t}{\partial p_t^i}\frac{\partial p_t^i}{\partial h_t^i} + \frac{\partial E_t}{\partial p_t^j}\frac{\partial p_t^j}{\partial h_t^i} + \frac{\partial E_t}{\partial p_t^k}\frac{\partial p_t^k}{\partial h_t^i} \\
&= (p_t^r - y_t^r) \times f'(p_t^r) \times -W_{hy}^i + (p_t^i - y_t^i) \times f'(p_t^i) \times W_{hy}^r \\
&\quad + (p_t^j - y_t^j) \times f'(p_t^j) \times W_{hy}^k + (p_t^k - y_t^k) \times f'(p_t^k) \times -W_{hy}^j.
\end{aligned}
\tag{56}
$$

$$
\begin{aligned}
\frac{\partial E_t}{\partial h_t^j} &= \frac{\partial E_t}{\partial p_t^r}\frac{\partial p_t^r}{\partial h_t^j} + \frac{\partial E_t}{\partial p_t^i}\frac{\partial p_t^i}{\partial h_t^j} + \frac{\partial E_t}{\partial p_t^j}\frac{\partial p_t^j}{\partial h_t^j} + \frac{\partial E_t}{\partial p_t^k}\frac{\partial p_t^k}{\partial h_t^j} \\
&= (p_t^r - y_t^r) \times f'(p_t^r) \times -W_{hy}^j + (p_t^i - y_t^i) \times f'(p_t^i) \times -W_{hy}^k \\
&\quad + (p_t^j - y_t^j) \times f'(p_t^j) \times W_{hy}^r + (p_t^k - y_t^k) \times f'(p_t^k) \times W_{hy}^i.
\end{aligned}
\tag{57}
$$

$$
\begin{aligned}
\frac{\partial E_t}{\partial h_t^k} &= \frac{\partial E_t}{\partial p_t^r}\frac{\partial p_t^r}{\partial h_t^k} + \frac{\partial E_t}{\partial p_t^i}\frac{\partial p_t^i}{\partial h_t^k} + \frac{\partial E_t}{\partial p_t^j}\frac{\partial p_t^j}{\partial h_t^k} + \frac{\partial E_t}{\partial p_t^k}\frac{\partial p_t^k}{\partial h_t^k} \\
&= (p_t^r - y_t^r) \times f'(p_t^r) \times -W_{hy}^k + (p_t^i - y_t^i) \times f'(p_t^i) \times W_{hy}^j \\
&\quad + (p_t^j - y_t^j) \times f'(p_t^j) \times -W_{hy}^i + (p_t^k - y_t^k) \times f'(p_t^k) \times W_{hy}^r.
\end{aligned}
\tag{58}
$$

Then,

$$
\begin{bmatrix}
\frac{\partial h_{r,m}}{\partial W_{hh}^r} = h_{r,t-1} & \frac{\partial h_{i,m}}{\partial W_{hh}^r} = h_{i,t-1} & \frac{\partial h_{j,m}}{\partial W_{hh}^r} = h_{j,t-1} & \frac{\partial h_{k,m}}{\partial W_{hh}^r} = h_{k,t-1} \\
\frac{\partial h_{r,m}}{\partial W_{hh}^i} = -h_{i,t-1} & \frac{\partial h_{i,m}}{\partial W_{hh}^r} = h_{i,t-1} & \frac{\partial h_{j,m}}{\partial W_{hh}^r} = h_{j,t-1} & \frac{\partial h_{k,m}}{\partial W_{hh}^r} = h_{k,t-1} \\
\frac{\partial h_{r,m}}{\partial W_{hh}^j} = -h_{j,t-1} & \frac{\partial h_{i,m}}{\partial W_{hh}^j} = -h_{k,t-1} & \frac{\partial h_{j,m}}{\partial W_{hh}^j} = h_{r,t-1} & \frac{\partial h_{k,m}}{\partial W_{hh}^j} = h_{i,t-1} \\
\frac{\partial h_{r,m}}{\partial W_{hh}^k} = -h_{k,t-1} & \frac{\partial h_{i,m}}{\partial W_{hh}^k} = h_{j,t-1} & \frac{\partial h_{j,m}}{\partial W_{hh}^k} = -h_{i,t-1} & \frac{\partial h_{k,m}}{\partial W_{hh}^k} = h_{r,t-1}
\end{bmatrix} = h_t^*.
\tag{59}
$$

The remaining terms $\frac{\partial h_t^r}{\partial h_m^r}, \frac{\partial h_t^i}{\partial h_m^i}, \frac{\partial h_t^j}{\partial h_m^j}$ and $\frac{\partial h_t^k}{\partial h_m^k}$ are recursive and are written as:

$$
\begin{aligned}
\frac{\partial h_{r,t}}{\partial h_{r,m}} = \prod_{n=m+1}^{t} & \frac{\partial h_{r,n}}{\partial h_{r,n}^{preact}}\frac{\partial h_{r,n}^{preact}}{\partial h_{r,n-1}} + \frac{\partial h_{r,n}}{\partial h_{i,n}^{preact}}\frac{\partial h_{i,n}^{preact}}{\partial h_{r,n-1}} \\
& + \frac{\partial h_{r,n}}{\partial h_{j,n}^{preact}}\frac{\partial h_{j,n}^{preact}}{\partial h_{r,n-1}} + \frac{\partial h_{r,n}}{\partial h_{k,n}^{preact}}\frac{\partial h_{k,n}^{preact}}{\partial h_{r,n-1}},
\end{aligned}
\tag{60}
$$

simplified with,

$$
\begin{aligned}
\frac{\partial h_{r,t}}{\partial h_{r,m}} = \prod_{n=m+1}^{t} & \frac{\partial h_{r,n}}{\partial h_{r,n}^{preact}} \times W_{hh}^r + \frac{\partial h_{r,n}}{\partial h_{i,n}^{preact}} \times W_{hh}^i \\
& + \frac{\partial h_{r,n}}{\partial h_{j,n}^{preact}} \times W_{hh}^j + \frac{\partial h_{r,n}}{\partial h_{k,n}^{preact}} \times W_{hh}^k.
\end{aligned}
\tag{61}
$$

Consequently,

$$\frac{\partial h_{i,t}}{\partial h_{i,m}} = \prod_{n=m+1}^{t} \frac{\partial h_{i,n}}{\partial h_{r,n}^{preact}} \times -W_{hh}^i + \frac{\partial h_{i,n}}{\partial h_{i,n}^{preact}} \times W_{hh}^r$$
$$+ \frac{\partial h_{j,n}}{\partial h_{j,n}^{preact}} \times W_{hh}^k + \frac{\partial h_{i,n}}{\partial h_{k,n}^{preact}} \times -W_{hh}^j. \tag{62}$$

$$\frac{\partial h_{j,t}}{\partial h_{j,m}} = \prod_{n=m+1}^{t} \frac{\partial h_{j,n}}{\partial h_{r,n}^{preact}} \times -W_{hh}^j + \frac{\partial h_{j,n}}{\partial h_{i,n}^{preact}} \times -W_{hh}^k$$
$$+ \frac{\partial h_{j,n}}{\partial h_{j,n}^{preact}} \times W_{hh}^r + \frac{\partial h_{j,n}}{\partial h_{k,n}^{preact}} \times W_{hh}^i. \tag{63}$$

$$\frac{\partial h_{k,t}}{\partial h_{k,m}} = \prod_{n=m+1}^{t} \frac{\partial h_{k,n}}{\partial h_{r,n}^{preact}} \times -W_{hh}^k + \frac{\partial h_{k,n}}{\partial h_{i,n}^{preact}} \times W_{hh}^j$$
$$+ \frac{\partial h_{k,n}}{\partial h_{j,n}^{preact}} \times -W_{hh}^i + \frac{\partial h_{k,n}}{\partial h_{k,n}^{preact}} \times W_{hh}^r. \tag{64}$$

The same operations are performed for **i,j,k** in Eq. 68 and $\frac{\partial E_t}{\partial W_{hh}}$ can finally be expressed as:

$$\frac{\partial E_t}{\partial W_{hh}} = \sum_{m=0}^{t} (\prod_{n=m+1}^{t} \delta_n) \otimes h_{t-1}^*, \tag{65}$$

with,

$$\delta_n = \begin{cases} W_{hh}^* \otimes \delta_{n+1} \times \alpha'(h_n^{preact}) & \text{if } n \neq t \\ W_{hy}^* \otimes (p_n - y_n) \times \beta'(p_n^{preact}) & \text{else.} \end{cases} \tag{66}$$

### 6.3.4 INPUT WEIGHT MATRIX

$\frac{\partial E_t}{\partial W_{hx}}$ is computed in the exact same manner as $\frac{\partial E_t}{\partial W_{hh}}$.

$$\frac{\partial E}{\partial W_{hx}} = \sum_{t=0}^{N} \frac{\partial E_t}{\partial W_{hx}}. \tag{67}$$

And,

$$\frac{\partial E_t}{\partial W_{hx}} = \sum_{m=0}^{t} \frac{\partial E_m}{\partial W_{hx}^r} + i\frac{\partial E_m}{\partial W_{hx}^r} + j\frac{\partial E_m}{\partial W_{hx}^i} + k\frac{\partial E_m}{\partial W_{hx}^k}. \tag{68}$$

Therefore $\frac{\partial E_t}{\partial W_{hx}}$ is easily extent as:

$$\frac{\partial E_t}{\partial W_{hx}} = \sum_{m=0}^{t} (\prod_{n=m+1}^{t} \delta_n) \otimes x_t^*. \tag{69}$$

### 6.3.5 HIDDEN BIASES

$\frac{\partial E_t}{\partial B_h}$ can easily be extended to:

$$\frac{\partial E}{\partial B_h} = \sum_{t=0}^{N} \frac{\partial E_t}{\partial B_h}. \tag{70}$$

And,

$$\frac{\partial E_t}{\partial B_h} = \sum_{m=0}^{t} \frac{\partial E_m}{\partial B_h^r} + i\frac{\partial E_m}{\partial B_h^r} + j\frac{\partial E_m}{\partial B_h^i} + k\frac{\partial E_m}{\partial B_h^k}. \tag{71}$$

Nonetheless, since biases are not connected to any inputs or hidden states, the matrix of derivatives defined in Eq. 59 becomes a matrix of 1. Consequently $\frac{\partial E_t}{\partial B_h}$ can be summarized as:

$$\frac{\partial E_t}{\partial B_h} = \sum_{m=0}^{t} \left( \prod_{n=m+1}^{t} \delta_n \right). \tag{72}$$

