# OpenReview forum: "Quaternion Recurrent Neural Networks"
_ICLR.cc/2019/Conference_

### Official Review · AnonReviewer3 · 2018-10-29
**Simple but nice development of the framework based on mostly well-known algebra but lacks experimental validation**

**Rating:** 7
**Confidence:** 5

**Review:**

After the discussion with authors, I am happy to recommend acceptance.
————————————————————

1.	In “Consequently, for each input vector of size N, output vector of size M, dimensions are split into four parts: the first one equals to r, the second is xi, the third one equals to yj, and the last one to zk to compose a quaternion Q = r1 + xi + yj + zk”, are you splitting dimension M or M\times N? And if you split M \times N (I believe that’s what you are doing), in which order you are splitting (row major right?) Please explain.
2.	I did not understand why authors didn’t go in the negative direction of the gradient in Eq. (10-11)?
3.	In section 3.4, authors mentioned “Moreover, an hyper-complex parameter cannot be simply initialized randomly and component-wise, due to the interactions between components.” which I strongly agree. But in Eq. (7) and (9) why the update rules and activation function are applied component wise?
4.	I really like the elegance in the parameter initialization. Couple of minor things here: (1) It’s better to mention in Eq. (16) why E(|W|) is 0 because of symmetry. (2) Reference should be 6.1 instead of 5.1.
5.	Another reasonable baseline will be using a complex network like (https://openreview.net/forum?id=H1T2hmZAb) and use the first two terms in Eq. (19) for representation. This will also possibly justify the usefulness of using higher order partials.
6.	The authors mentioned multiple times about the achieved state-of-the-art results without giving any citation. As a reader not well versed in the acoustic domain, it will be nice to see some references to cross-validate the claim made.



General Comments:
1.	I understand the necessity of defining RNN/ LSTM model in the space of quaternions. But unit quaternions can be identified with other spaces where convolution is defined recently, e.g., with S^3 (https://arxiv.org/abs/1809.06211). I can see that this paper is contemporary, but at least can authors comment on the applicability of this general method in their case? Given that in NIPS’18 the following paper talked about RNN model on non-Euclidean spaces (https://arxiv.org/pdf/1805.11204.pdf), one can extend these ideas to develop an RNN model in the space of quaternions. Authors should look into it rigorously as future directions? But at least please comments on the applicability.
2.	The experimental results section is somewhat weak, the overall claim of using fewer parameters and achieving comparable results is only validated on TIMIT data. More experimentation is necessary.
3.	In terms of technical novelty, though quaternion algebra is well-known, I like the parameter initialization algorithm. I can see the merit of this in ML/ vision community.

Pros:
1. Nice well grounded methodological development on well-known algebra. (simple but elegant, so that's good).
2. Nicely written and all the maths check out (that's good).
3. Experimental result on TIMIT dataset shows usefulness in terms of using fewer parameters (but still can achieve SOA results).

Cons:
1. See my comments above. I expect the authors to rebut/ address the aforementioned comments. Overall though simple but nice (and necessary) development of RNN/ LSTM framework in the space of quaternions.
2. Lacks extensive experimental validation.

My reason for my rating is mainly because of (1) lack of experimental validation. (2) being aware of the recent development of general RNN model on non-Euclidean spaces, I want some comments in this direction (see detailed comment and reference above).

---

> ### Author Response · Authors · 2018-11-09
> **Thanks you + answers [2/2]**
>
> ——— General comments
>
> 1. This is a critical point, indeed. The question « Complex and quaternion valued networks are ok, but what if we want to go in higher dimensions ? » is a common question. We are aware that many works exist in higher dimensions (Octonions, sedenions ..) or even on « generic » models that could apply to any dimensionality (like Clifford algebra based neural networks, or ManifoldNets). Therefore, we acknowledge the fact that it should be possible to reduce the dimension of such different spaces to 4, but then, we would end up with neural networks that will behave as quaternion-valued neural networks. Quaternion-valued neural networks are a special case of such high-dimensional algebras, and are thus suitable to perfectly solve specific problematics that could be very useful for many domain areas, such as image processing, human pose estimation, 3D transformations,…,. It is therefore important to first clearly define a quaternion-valued neural network as a specific neural network using a particular algebra. However, it is clear that there is a need for higher dimensional neural networks (such as ManifoldNets), and there is plenty of rooms for investigations.
>
> 2. We agree that TIMIT is too small. Therefore, and as described in the answer to questions of  reviewer 2, we added results on a larger speech recognition task (Wall Street Journal) in the supplementary materials. Reported results confirm those observed on the TIMIT task with fewer parameters and slightly lower Word Error Rates.
>
> 3. We thanks reviewer 3 for this encouraging statement.
>
> We truly hope that we answered all your questions and remarks, and we are still open to any discussion on this work.

---

> ### Author Response · Authors · 2018-11-09
> **Thanks you + answers[1/2]**
>
> We would like to first thank reviewer 3 for the detailed and useful feedback. We start by addressing each one of the initial point raised:
>
> 1. Let us take the example of an input vector X of size N=256 and an output vector O of size M=512. During computations, both X and O are one-dimensional real-valued vectors. Numbers contained in X[0,…,63] are real components, while X[64,…,127] belong to the component i. Therefore the first quaternion is X[0] + X[64]i + X[128]j + X[192]k, and the same considerations can be done for the output vector O. At the end, we have N/4 input quaternions and M/4 output quaternions.
>
> 2. The update phase of the NN parameters with respect to the gradient direction is actually depending on the task. We corrected it to go in the negative direction of the gradient, we thank reviewer 2 for suggesting this correction.
>
> 3.The authors agree with the fact that the split activation functions does not seem to perfectly suit quaternion networks. Therefore we added a statement in the paper to motivate the use of split activation functions. Nonetheless, these quaternion activation functions have been found to be more stable (purely quaternion functions have singularities), and easier to compute, making them interesting for QRNNs. We plan in a near future to investigate QNNs that will use quaternions from the input to the output (pure quaternion activations, full rotations), but we believe that these networks might be harder to train due to singularities deriving from the use of quaternion algebra. Furthemore, the BPTT for quaternions is defined based on the initial work of back propagation in the quaternial domain proposed by P. Arena. In the latter, the loss (Eq. 9) has to be calculated with respect to each component of the quaternion. Indeed, we have to evaluate how much each component of a given quaternion parameter affects the whole loss. Then by applying the chain rule, we end up with a component-wise product in Eq. 13 due to the split activations, which is simpler and way less computationally intensive than calculating the derivative of a quaternion-valued function.
>
> 4. Fixed.
>
> 5. We thought a lot about baselines during the experiments. The main issue is that it is not possible to compare complex-valued NN (CVNNs) to QNNs in a fair setting. Indeed, in the case proposed by reviewer 2, CVNNs will clearly have less information and will give worst results. Then we could use CVNNs with magnitude and phase directly from the signal, but the input space would be different compared to QNN, and we won’t have comparable results. Many papers use well engineered features, more-complex structures (attention mechanisms, gates,..,), or even training regularization (Batch-normalization), and it won’t be fair to compare our « vanilla » QRNN and QLSTM to such models. For these reasons, we have decided to add real-valued RNN and LSTM, that are exactly the same than quaternion-valued ones, to obtain fair comparisons. However, it is clear that it could be interesting to build and investigate a complex and state-of-the art quaternion-valued model, but we have to  first introduce the basics of QRNN-QLSTM based models.
>
> 6. The authors agree with this remark. As suggested by reviewer 3, we added more citations and written results of the literature in the paper (in term of PER) to help the reader to better compare and evaluate the observed results.

---

### Official Review · AnonReviewer1 · 2018-11-02
**Quaternion Recurrent Neural Networks**

**Rating:** 7
**Confidence:** 5

**Review:**

Quality: sufficient though there are issues. Work done in automatic speech recognition on numerous variants of recurrent models, such as interleaved TDNN and LSTM (Peddinti 2017), is completely ignored [addressed in the revision]. The description of derivatives needs to mention the linear relationship between input features and derivatives (see trajectory HMMs by Zen and Tokuda) [addressed in the revision]. TIMIT is a very simple task [addressed by adding WSJ experiments]. Derivations in the appendices could be connected better [addressed in the revision].

Clarity: sufficient. It would be good to see some discussion of 1) split activations and other possible options [short comment added in the revision] if any 2) expressions of derivatives and their connection to standard RNN derivatives [short comment added in the revision], 3) computational complexity [addressed in the revision].

Originality: sufficient. This paper describes the extension of quaternion feed-forward neural networks to recurrent neural networks and a parameter initialisation method in the quaternial domain.

Significance: sufficient.

Pros: Audience interested in quaternial neural networks would benefit from this publication. Experimental results even if limited suggest that quaternial representation may offer a significant reduction in the number of model parameters at no loss in performance.

Cons: The choice of derivatives to yield quaternions as there are other more interesting views to contemplate both in speech and other fields. A simple task makes it hard to judge how the quaternion extension would scale.

Other:

The format of references, the use of a number in parentheses, is unusual and distractive. [fixed in the revision]
Please at least name all the terms used in the main paper body even if they are defined later in the appendix (e.g. h_{t}^{*} in equation 10). [fixed in the revision]
Do both W_{hh} and b_{h} contain the same \delta_{hh}^{t} term in their update equation 11? [fixed in the revision]
Page 7 by mistake mentions 18.2% which cannot be found in the Table 1. [fixed in the revision]
Page 12 "is equals to" [remains in the revision]

---

> ### Author Response · Authors · 2018-11-09
> **Thanks you + answers**
>
> We would like to first thank reviewer 1 for the useful feedback. In the following, we address typos and general comments.
>
> ——— Typos and general comments
>
> The format of references has been modified to match the standard of the ICLR format (Name et al, year).
>
> The authors agree with the fact that the notation have to be better explained to make the paper more clear for the reader. Therefore, we have added a sentence to clarify h_{t}^{*}.
>
> We would like to thank the reviewer 1 to highlight that b_{h} was given the wrong delta during the backpropagation. The right equation for b_{h} has been added to the paper as well as in the supplementary material.
>
> Others typos have been corrected.
>
> ——— Quality
>
> As suggested by reviewer 1, we added the missing references of prior works on ASR systems in the introduction, and the linear relation between input features and derivatives.
>
> The authors agree with fact that TIMIT is a very simple task,  but this framework allows us to evaluate the relevance of RNNs in terms of performance and the number of parameters required. Therefore, in the revised version of the paper, we have added experiments on the Wall Street Journal (WSJ) speech recognition task (in the supplementary materials) based on both the 14 and the 81 hours training data-sets. Experiments are conducted with the same configurations than the ones from the models that have obtained the best results observed during the experiments on speech recognition on the TIMIT data-set. As expected, QLSTMs scale well (such as for real-valued LSTM) to larger data-sets, and the performances observed during these experiments support the fact that QLSTMs perform better (w.r.t WER), and  with fewer parameters..
>
> ——— Clarity
>
> Quaternion NNs suffer from the fact of being still little employed. Therefore, we understand that many concepts such as the « split activation functions » raise legitimate questions. We added some words on the paper to motivate the use of split activation functions. Nevertheless, and as we have mentioned throughout new citations, the split activation functions have already been well investigated, and we could only paraphrase what original authors demonstrated. However, this point raised by Reviewer 1 have to be investigated during a dedicated work, to allow the reader to easily follow the study that will compare different activation functions as well as different function methods (split or not).
>
> The authors agree with reviewer 1 and, therefore,  we have added a paragraph to clarify the decomposition of derivatives (Below Eq. 9) in the paper and in the appendix (Below Eq. 40). Indeed, the derivatives of different elements/views of a same feature with a real valued BPTT process, do not allow the RNN based model to learn how to compute the whole dynamic of the error (dE/dW), due to the fact that the dynamic of each element composing the features, are not merged/mixed to compose the derivative of the whole error observed. This process of merging partial derivatives from each elements (r, i, j, k for quaternions and 4 different input features for real numbers) is managed by the weight matrices and hidden states in the context of real-valued RNN based models during the learning process. The author agree that these intuitions have to be supported by solid experiments and model analyses. Therefore, we also plan to investigate the internal dynamic (through partial derivatives that contribute to the total dynamic) of QRNNs compared to real-valued RNNs ones to better understand the benefits from the QBPTT (in addition to better results and less parameters).
>
> Computational complexity is also a very good point, but hard to fairly answer in the current state of QNNs. Nonetheless, and as requested by reviewer 1, we added a paragraph to the paper (Appendix 6.1.2)), to acknowledge the fact that computational complexity can be a problem. From a pure computational complexity perspective: QLSTM = LSTM = O(n^2). Indeed, due to the real-valued representation of quaternions, QLSTMs perform the same matrices operations, but with 4 times bigger matrices. From a computational time perspective, a simple forward propagation between two quaternion neurons involves 28 computations. Therefore QNNs are slower to train (2 to 3 times slower, depending on the model) due to this much higher number operations. Nonetheless, we also know that such computations are matrices products. We believe that a proper GPU engineering (cuDNN kernel) of the Hamilton product could drastically reduce the computation time by doing these 28 computations in parallel, implying a more efficient usage of the available resources. Furthermore, with a proper cuDNN kernel, one will obtain a better memory / computation rate. Indeed, QNNs are doing more computations, but with fewer parameters. This point will be detailed in a proper section in the appendix of the final version of the paper.

---

> > ### Author Response · Authors · 2018-11-09
> > **Thanks you + answers [2/2]**
> >
> > ——— Pros and Cons
> >
> > As stated above, we added WSJ experiments to validate the results observed with the small TIMIT dataset. Reviewer 1’s statement about actual quaternion acoustic feature is definitely true, and we propose in the conclusion to investigate novel multi-view features that could be better adapted.
> >
> > We truly hope that we answered all the remarks and questions of reviewer 1, and we are available for any further discussion.

---

### Official Review · AnonReviewer2 · 2018-11-06
**Explores a good and important direction, with encouraging results**

**Rating:** 8
**Confidence:** 4

**Review:**

The paper takes a good step toward developing more structured representations by exploring the use of quaternions in recurrent neural networks.  The idea is motivated by the observation that in many cases there are local relationships among elements of a vector that should be explicitly represented.  This is also the idea behind capsules - to have each "unit" output a vector of parameters to be operated upon rather than a single number.   Here the authors show that by incorporating quaternions into the representations used by RNNs or LSTMs, one achieves better performance at speech recognition tasks using fewer parameters.

The quaternionic representation of the spectrogram chosen here seems a bit arbitrary.  Why are these the attributes to be packaged together?  its not obvious.  Shouldn't this be learned?

---

> ### Author Response · Authors · 2018-11-09
> **Thanks you + further informations**
>
> The authors thank the reviewer for the positive and constructive feedback. We appreciate that the reviewer finds that our paper on QRNN is clearly explained, viable and thoroughly evaluated.
>
> In this work, we decided to show that even with traditional acoustic features (Mel-filter-bank + derivatives), we could motivate and introduce quaternion-valued recurrent neural networks. Nonetheless, as underlined by reviewer 2, a future work will be to investigate proper quaternion acoustic features (or even other domains features). Indeed, current features are mostly engineered for a real-valued representation, and there is plenty of rooms to explore quaternion-valued features (such as complex-valued features in the case of speech recognition with complex neural networks, or quaternion Fourier Transform).

---

### Meta-Review · Area_Chair1 · 2018-12-14
**Interesting work applying quaternion representations to neural networks**

**Confidence:** 4
**Recommendation:** Accept (Poster)

**Metareview:**

The authors derive and experiment with quaternion-based recurrent neural networks, and demonstrate their effectiveness on speech recognition tasks (TIMIT and WSJ), where the authors demonstrate that the proposed models can achieve the same accuracy with fewer parameters than conventional models. The reviewers were unanimous in recommending that the paper be accepted.